# Derivation of the Omega-3 Index from EPA and DHA Analysis of Dried Blood Spots from Dogs and Cats

**DOI:** 10.3390/vetsci10010013

**Published:** 2022-12-26

**Authors:** William S. Harris, Kristina H. Jackson, Heather Carlson, Nils Hoem, Tonje E. Dominguez, Lena Burri

**Affiliations:** 1Fatty Acid Research Institute, Sioux Falls, SD 57106, USA; 2Department of Internal Medicine, Sanford School of Medicine, University of South Dakota, Sioux Falls, SD 57105, USA; kristina@omegaquant.com; 3OmegaQuant Analytics, LLC., Sioux Falls, SD 57106, USA; 4All-City Pet Care Veterinary Emergency Hospital, Sioux Falls, SD 57105, USA; heatherintea@hotmail.com; 5Aker BioMarine Antarctic AS, NO-1327 Lysaker, Norway; nils.hoem@akerbiomarine.com (N.H.); tonje.dominguez@akerbiomarine.com (T.E.D.); lena.burri@akerbiomarine.com (L.B.)

**Keywords:** omega-3 index, eicosapentaenoic acid, docosahexaenoic acid, omega-3 fatty acids, erythrocytes, biomarkers, fish oil, dogs, cats

## Abstract

**Simple Summary:**

Omega-3 fatty acids have known health benefits for humans, but the extent to which these benefits extend to animals is still being discovered. Any benefits derived depend on achieving specific blood omega-3 levels, and the metric often used in human studies is the Omega-3 Index (O3I). It is a valid measure of omega-3 status reflecting the eicosapentaenoic acid (EPA) and docosahexaenoic acid (DHA) content of red blood cell membranes. The O3I is often calculated from an analysis of whole blood dried on filter paper; and how to convert whole blood EPA + DHA into the O3I is well established in humans, but not yet for other species. This study analyzed blood samples from dogs and cats and developed equations to estimate the O3I in both of these species. Having a validated measure of omega-3 status in these animals will facilitate further research to define exactly what blood levels of omega-3 fatty acids are optimal for each species. These levels can then be used by veterinarians and pet owners alike to gauge the amount of supplemental EPA and DHA each animal needs to achieve an optimal O3I.

**Abstract:**

The Omega-3 Index (O3I) is the red blood cell (RBC) eicosapentaenoic acid (EPA) plus docosahexaenoic acid (DHA) content expressed as a percentage of total RBC fatty acids. Although a validated biomarker of omega-3 status in humans, little is known about the O3I status of dogs and cats; species in which omega-3 fatty acids have known health benefits. The purpose of this study was to develop equations to predict the O3I in these species from a dried blood spot (DBS) analysis. Random blood samples from 33 dogs and 10 cats were obtained from a community veterinary clinic. DBS and RBC samples were analyzed for fatty acid composition. For both species, the R^2^ between the DBS EPA + DHA value and the O3I was >0.96 (*p* < 0.001). The O3I was roughly 75% lower in dogs and cats than in humans. We conclude that the O3I can be estimated from a DBS sample, and the convenience of DBS collection should facilitate omega-3 research in these companion animals.

## 1. Introduction

The Omega-3 Index (O3I) is the red blood cell (RBC) eicosapentaenoic acid (EPA) plus docosahexaenoic acid (DHA) content expressed as a percentage of total RBC fatty acids [1]. It has been validated as a biomarker of omega-3 status in humans [2] and correlates linearly with EPA + DHA content of multiple tissues [3,4,5]. Being an RBC and not a plasma-based metric, the O3I has lower within-person variability [6] and is largely unperturbed by an acute intake of omega-3 fatty acids [7], two features that make it a metric of potentially greater clinical utility than plasma EPA + DHA measurements. In this regard, the O3I is analogous to hemoglobin A1c which is a more stable, long-term marker of glycemic status than is plasma glucose. The O3I also has predictive power vis à vis a variety of clinically relevant endpoints such as cardiovascular disease [8], dementia [9], and death from any cause [10]. In these settings, targeting an O3I of ≥8% (for humans) appears to afford the best protection [11]. Although an RBC metric, the O3I can be accurately calculated from a whole blood sample, whether obtained from a blood tube or as a dried blood spot (DBS) [12] (Data supporting this claim are presented in Appendix A).

The EPA + DHA content of whole blood (whether liquid or dried) is different from that of the RBCs or of the accompanying plasma, since whole blood is a mixture of both with each compartment having a different fatty acid profile [13]. In humans, several studies have been published in which the O3I was computed from the EPA + DHA content of either DBS or plasma samples [1,14,15]. 

Most of the previous studies examining the health benefits of a higher vs. lower O3I have been done in humans, but there is a growing interest in quantifying the omega-3 status in animals as well, since omega-3 supplementation has been shown to have beneficial effects on a variety of health indices in dogs [16,17,18,19] and cats [20,21,22]. Access to information about blood (and thus systemic) omega-3 status in these species could help veterinarians titrate these levels using omega-3 supplements or enriched foods and thereby achieve said health benefits. Although blood sampling for these species typically requires phlebotomy, DBS samples require only a drop of blood which could potentially be collected without phlebotomy, making home collection possible. In addition, DBS cards are safer, simpler and cheaper to transport for laboratory testing because they can be sent at ambient temperature. Therefore, knowing how to determine the O3I from a DBS sample in these companion animals will have clinical utility.

The primary purpose of this study was to derive equations to convert DBS EPA + DHA levels into the O3I in these two species. The secondary purpose was to do the same starting with plasma EPA + DHA levels as this has been the most commonly reported measure of omega-3 status in veterinary research. To have equations that can translate published plasma EPA + DHA levels into the O3I could help to harmonize earlier studies around a single omega-3 status marker. Human samples were included in these studies as a point of reference.

## 2. Materials and Methods

### 2.1. Animals

The dog and cat samples used for this analysis were left over from routine care of animals being seen at the All-City Pet Care Veterinary Emergency Hospital (Sioux Falls, SD, USA). No dietary or demographic information about the animals was collected because the purpose of the study, as noted above, was simply to compare the EPA + DHA levels in different blood compartments; therefore, understanding why levels differed from animal to animal was not relevant to this investigation. The human data were from randomly chosen, deidentified, blood samples arriving at OmegaQuant Analytics (Sioux Falls, SD, USA) for routine fatty acid testing.

### 2.2. Laboratory

The EPA + DHA content for RBCs [23], DBS [12] and plasma samples [24] (as a percent of total fatty acids) was determined by gas chromatography using flame ionization detection as previously described. For plasma analyses, absolute EPA + DHA concentrations were determined by including an internal standard (tri-tricosanoic acid).

### 2.3. Statistical

Means, standard deviations and regression analyses were calculated using Excel (Microsoft 365). 

## 3. Results

For dogs the random collection of samples from a community veterinary hospital provided a wide range of omega-3 levels, presumably because some owners were feeding their pets omega-3 enriched foods and/or supplements. Considerably fewer samples were available for cats for reasons described in the Discussion. For these two species, comparisons of DBS EPA + DHA with the RBC-based O3I metric produced the conversion equations shown in Table 1 and illustrated in Figure 1 and Figure 2. The equation for human samples is included as a comparator (Table 1 and Figure 3). The relationships between the RBC and DBS metrics were linear in all species. All comparisons had R^2^ values > 0.88 (i.e., r > 0.93), and all were statistically significant (*p* < 0.001). It is clear from Figure 1 and Figure 2 that O3I values for unsupplemented animals can be quite low (~0.5% EPA + DHA) compared with humans (~2% EPA + DHA).

The O3I calculated from a DBS EPA + DHA value of 4% (as an example), varied by species, being highest for humans and lowest for dogs and cats (Table 1). In addition, the ranges of O3I values from the animals included in this report also varied. For dogs, the low to high O3I spanned about a 20-fold range; for cats, about an 8-fold range; and for humans, 6-fold (Table 1). 

Relations between the O3I and plasma EPA + DHA [the latter expressed both as a percent of total plasma fatty acids and/or as a concentration (ug/mL)] for three of the species are shown in Appendix A. In the plasma comparisons with DBS, stronger correlations are always observed when expressing plasma fatty acid levels as percentages of total fatty acids than as concentrations. The full 24-fatty acid profile from RBCs, DBS and plasma (the latter as both percent composition and concentrations) are shown in Table 2, Table 3 and Table 4 for dogs, cats and humans, respectively. 

## 4. Discussion

The purpose of this study was to develop equations that could be used to convert DBS omega-3 data into the RBC-based metric, the O3I, in two companion animal species: dogs and cats, as well as in humans. The data available to us were obtained from a community veterinary practice where pet dogs and cats are brought for routine care. Because blood is more commonly drawn on dogs than cats, we obtained only one-third as many samples from the latter compared with the former. (The reason for this is that dogs are annually tested for heartworm using an in-house blood test. However, for cats, it is not standard of practice to test for heartworm, and the test is not available for in-clinic analysis but requires shipment to outside labs. Hence, cat blood is only drawn in surgical cases, not in annual check ups.) Because we have considerably less data in cats, we have less confidence in their equation compared to that from dogs, and further studies are clearly needed to confirm especially the cat equation. 

As noted above, we assumed that blood levels of EPA + DHA were at steady state for all three species studied here as blood samples were collected at random. Interestingly, the equations to convert DBS EPA + DHA to RBC O3I were relatively similar for dogs and cats, but these differed substandially from the equation derived from humans. For example, a DBS EPA + DHA value of 4% would translate into an O3I of 5.9% for humans, but 2.9% for dogs and cats. The DBS values are lower than the RBC values in humans but higher in the animals. This is because, as noted above, DBS (whole blood) samples are a mix of plasma and cells, and since the plasma percent EPA + DHA is higher than the RBC EPA + DHA in the animals (vice versa in humans), the DBS percent values are higher than the RBC values. The extent to which this is a coincidence or not, and for how many animal species it would hold, is unknown. Another consequence of blood being an approximately 55:45 mix of plasma and cells (for all three species [25]) is that whole blood levels of virtually all fatty acids, not just EPA and DHA, are approximately the average of the RBC and plasma values. 

Why dogs and cats would have a lower O3I than humans is not immediately clear. Perhaps domestication and feeding them grain-based food (instead of raw meat, as they would consume in the natural state) may be at least part of the explanation. However, just eating raw meat would not raise the O3I unless it was oily fish (e.g., salmon, mackerel, sardines, etc.).

The rather wide distribution of omega-3 levels in the dogs suggests a variety of intakes of EPA and DHA from different kinds of dog foods. The same applies to cats who were, at one time, believed to be obligate carnivores owing to their reported inability to convert the 18-carbon essential fatty acids, linoleic and alpha-linolenic acids, to their longer-chain metabolites (i.e., arachidonic acid and EPA + DHA) [26]. However, when more sensitive methods were applied, this belief was shown not to be the case [27]. Thus, the EPA and DHA levels in cats may be driven both by synthesis from alpha-linolenic acid and by direct consumption of the marine omega-3 fatty acids.

This study had strengths and limitations. Among the former are the use on a single laboratory methodology to derive fatty acid levels from multiple sample types. This may also be viewed a limitation to the generalized translation of omega-3 levels in one blood compartment to another using other laboratory methods. As to limitations, we have already aluded to the rather small sample size for cats. In addition, we had no information regarding the sex and age of the animals, nor any data on current intake of fatty acids. We also excluded from this study plasma samples that were visibly lipemic, reflecting the recent consumption of a high fat meal. Since chylomicrons are very rich in triglycerides (i.e., FAs), the DBS omega-3 results would be artificially lower and less translateable to RBC EPA + DHA levels. Hence, our equations should only be used in non-lipemic samples (i.e., those drawn >8 h after a high fat meal). A final limitation would be our inability (due to limited resources) to repeat the study with a validation set to test the derived equations in a new group of animals. 

## 5. Conclusions

In this study we analyzed three blood sample types (RBCs, plasma, whole blood) in three species (dogs, cats and humans) in order to create equations that could be used to estimate the RBC EPA + DHA% (i.e., the O3I) from the EPA + DHA% in the other two sample types, in particular from the DBS samples. An important finding was that the equation used for humans cannot be used (accurately) in dogs and cats. Future studies in other animal species will be needed to define this conversion equation for those species. 

Having a method to make these conversions could help facilitate future studies that seek to define the optimal range of the O3I for dogs and cats similar to what has been done in humans, where risk (for cardiovascular disease) is highest at <4%, intermediate between 4 to 8%, and low above 8% [11]. It seems likely that optimal or target levels will be species-specific (and possibly disease-specific as well since dogs and cats are more prone to chronic inflammatory conditions than to cardiovascular disease which is more common in humans). The previously mentioned favorable health effects of omega-3 supplementation in dogs and cats (which necessarily raises the O3I, but to currently unknown levels) suggest that this will be the case.

## Figures and Tables

**Figure 1 vetsci-10-00013-f001:**
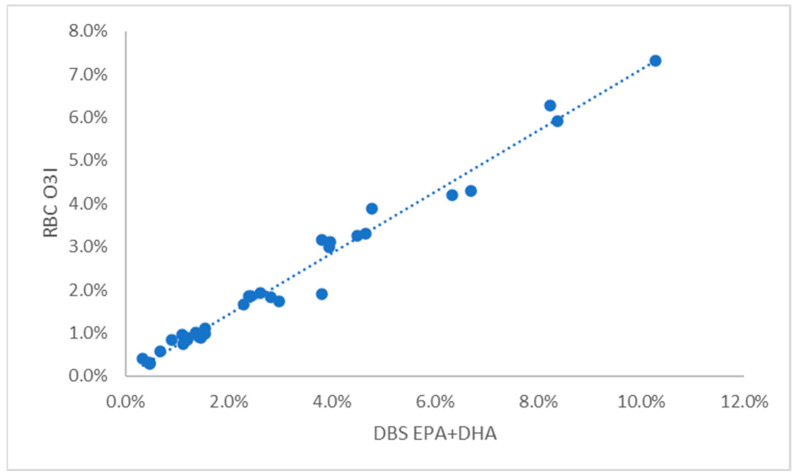
Relationship between the EPA + DHA content of DBS samples and the Omega-3 Index in 33 dogs. DBS, dried blood spot; RBC, red blood cell.

**Figure 2 vetsci-10-00013-f002:**
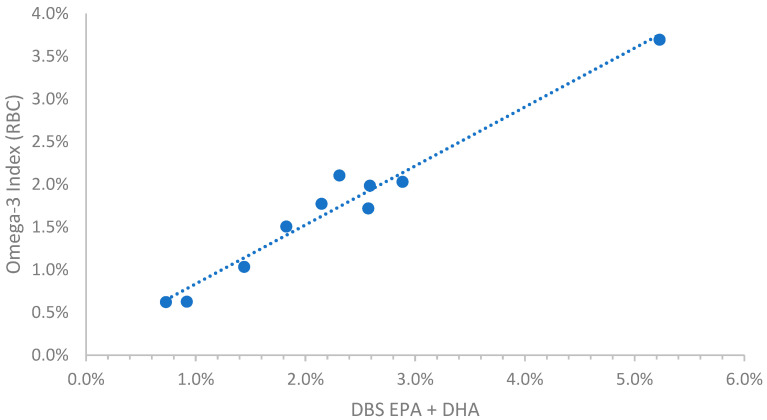
Relationship between the EPA + DHA content of DBS samples and the Omega-3 Index in 10 cats. DBS, dried blood spot; RBC, red blood cell.

**Figure 3 vetsci-10-00013-f003:**
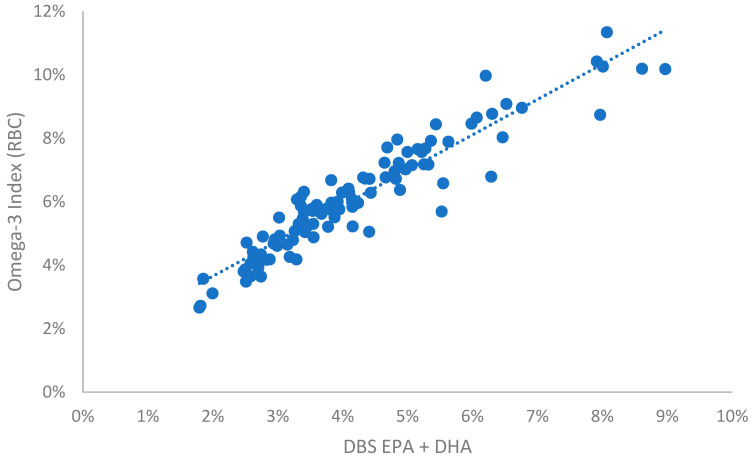
Relationship between the EPA + DHA content of DBS samples and the Omega-3 Index in 100 random human blood samples.

**Table 1 vetsci-10-00013-t001:** Equations to convert DBS EPA + DHA (x) into the Omega-3 Index (RBC EPA + DHA %) (y) by species.

Species	Equation	R^2^	O3I Equivalent of a DBS EPA + DHA of 4%	Approximate Range of O3I Values
Dog (*n* = 33)	y = 0.709x + 0.0002	0.98	2.9%	0.3 to 7%
Cat (*n* = 10)	y = 0.690x + 0.0014	0.96	2.9%	0.5 to 4%
Human (*n* = 100)	y = 1.114x + 0.014	0.88	5.9%	2 to 12%

**Table 2 vetsci-10-00013-t002:** Fatty acid composition of red blood cells, dried blood spots, and plasma (both as a percent of total fatty acids and as concentrations) from 33 dogs.

	RBC %		DBS %		Plasma %		Plasma (ug/mL)	
	Mean	SD	Mean	SD	Mean	SD	Mean	SD
C14:0	0.11%	0.04%	0.25%	0.11%	0.31%	0.12%	17.1	8.0
C16:0	14.85%	1.63%	13.60%	1.71%	13.05%	1.95%	718.9	207.1
C16:1n7t	0.10%	0.04%	0.14%	0.05%	0.13%	0.05%	6.9	2.9
C16:1n7	0.35%	0.17%	0.92%	0.47%	1.14%	0.51%	64.2	38.3
C18:0	29.23%	1.56%	20.34%	1.71%	16.85%	1.84%	946.0	330.6
C18:1t	0.64%	0.30%	0.59%	0.31%	0.55%	0.31%	30.0	18.7
C18:1n9	11.36%	1.23%	14.65%	2.90%	15.78%	3.65%	873.9	315.8
C18:2n6t	0.19%	0.05%	0.28%	0.08%	0.20%	0.07%	10.7	3.2
C18:2n6	10.83%	1.49%	21.04%	2.82%	27.16%	2.78%	1511.7	467.9
C20:0	0.07%	0.02%	0.11%	0.04%	0.09%	0.04%	5.1	2.5
C18:3n6	0.05%	0.02%	0.21%	0.06%	0.31%	0.09%	17.0	6.5
C20:1n9	0.45%	0.10%	0.33%	0.11%	0.17%	0.09%	8.9	4.4
C18:3n3	0.19%	0.16%	0.57%	0.46%	0.81%	0.68%	47.8	48.8
C20:2n6	0.29%	0.07%	0.20%	0.05%	0.23%	0.08%	12.5	5.1
C22:0	0.07%	0.02%	0.21%	0.05%	0.08%	0.03%	4.1	1.7
C20:3n6	1.29%	0.24%	1.03%	0.24%	0.87%	0.29%	48.6	23.7
C20:4n6	24.74%	2.34%	19.46%	3.36%	16.27%	3.82%	914.5	355.9
C24:0	0.09%	0.03%	0.18%	0.05%	0.07%	0.03%	3.9	1.5
C20:5n3	1.14%	1.16%	1.51%	1.51%	1.77%	1.75%	98.6	101.2
C24:1n9	0.37%	0.07%	0.43%	0.11%	0.12%	0.05%	6.2	2.6
C22:4n6	1.37%	0.68%	0.85%	0.47%	0.48%	0.34%	25.7	17.2
C22:5n6	0.22%	0.08%	0.19%	0.07%	0.12%	0.06%	6.3	3.1
C22:5n3	0.99%	0.29%	1.38%	0.44%	1.63%	0.57%	91.4	49.4
C22:6n3	1.03%	0.74%	1.52%	1.15%	1.83%	1.41%	100.8	77.7
O3I/[FA]	2.17%	1.83%	-	-	-	-	5570.6	419.3

Fatty acids are designated as CX:YnZ where CX = carbon number, Y the number of double bonds, and Z the position of the first double bond counting from the methyl end of the molecule. The “*t*” refers to the double bond being in the *trans* configuration; all other double bonds are in the *cis* configuration.

**Table 3 vetsci-10-00013-t003:** Fatty acid composition of red blood cells, dried blood spots, and plasma (both as a percent of total fatty acids and as concentrations) from 10 cats.

	RBC		DBS		Plasma %		Plasma (ug/mL)	
	Mean	SD	Mean	SD	Mean	SD	Mean	SD
C14:0	0.14%	0.05%	0.18%	0.11%	0.24%	0.10%	8.3	4.6
C16:0	17.85%	1.26%	15.02%	1.26%	13.10%	1.86%	454.7	243.2
C16:1n7t	0.07%	0.04%	0.10%	0.05%	0.09%	0.04%	3.3	2.5
C16:1n7	0.32%	0.12%	0.73%	0.26%	1.01%	0.36%	30.5	8.3
C18:0	25.85%	1.04%	19.84%	1.18%	17.59%	1.74%	586.4	250.2
C18:1t	0.86%	0.58%	0.62%	0.34%	0.58%	0.35%	22.3	20.3
C18:1n9	10.34%	1.09%	15.34%	1.94%	18.51%	2.86%	646.7	373.2
C18:2n6t	0.17%	0.03%	0.19%	0.05%	0.26%	0.07%	8.1	2.9
C18:2n6	18.37%	2.95%	25.62%	3.59%	31.45%	4.86%	1066.6	503.0
C20:0	0.10%	0.03%	0.17%	0.06%	0.12%	0.05%	3.6	1.1
C18:3n6	0.05%	0.02%	0.10%	0.05%	0.15%	0.07%	4.4	1.2
C20:1n9	0.45%	0.09%	0.33%	0.10%	0.17%	0.07%	5.3	1.9
C18:3n3	0.33%	0.32%	0.55%	0.45%	0.76%	0.60%	21.8	18.0
C20:2n6	0.75%	0.28%	0.48%	0.23%	0.29%	0.16%	8.9	5.1
C22:0	0.08%	0.02%	0.28%	0.05%	0.09%	0.03%	2.7	0.8
C20:3n6	1.04%	0.14%	0.88%	0.17%	0.80%	0.29%	23.5	2.0
C20:4n6	18.94%	3.03%	14.88%	3.26%	10.88%	4.08%	359.3	188.4
C24:0	0.13%	0.03%	0.24%	0.04%	0.08%	0.02%	2.4	0.8
C20:5n3	0.67%	0.52%	0.70%	0.69%	0.69%	0.85%	18.2	18.5
C24:1n9	0.74%	0.14%	0.73%	0.13%	0.09%	0.02%	3.0	1.0
C22:4n6	1.18%	0.29%	0.76%	0.12%	0.32%	0.12%	10.1	4.3
C22:5n6	0.22%	0.07%	0.22%	0.11%	0.13%	0.06%	4.7	2.8
C22:5n3	0.32%	0.06%	0.45%	0.12%	0.53%	0.13%	18.7	12.2
C22:6n3	1.04%	0.44%	1.56%	0.77%	2.04%	1.04%	75.0	65.6
O3I / [FA]	1.71%	0.89%	-	-	-	-	3388.8	277.1

Fatty acid nomenclature defined in Table 2.

**Table 4 vetsci-10-00013-t004:** Fatty acid composition of red blood cells, dried blood spots, and plasma (both as a percent of total fatty acids and as concentrations) from 100 humans.

	RBC (%)		DBS (%)		Plasma (%)		Plasma (ug/mL)	
	Mean	SD	Mean	SD	Mean	SD	Mean	SD
C14:0	0.24%	0.11%	0.47%	0.19%	0.69%	0.28%	22.1	13
C16:0	23.18%	1.23%	22.43%	1.40%	21.05%	1.86%	655.3	181.7
C16:1n7t	0.10%	0.03%	0.12%	0.04%	0.13%	0.04%	4.1	1.4
C16:1n7	0.17%	0.10%	0.82%	0.40%	1.38%	0.60%	43.8	24.6
C18:0	18.16%	0.75%	12.17%	1.03%	6.57%	0.74%	201.5	45.3
C18:1t	0.49%	0.08%	0.41%	0.11%	0.32%	0.13%	10	5.1
C18:1n9	14.44%	1.03%	18.22%	2.18%	21.50%	3.07%	670.7	209.8
C18:2n6t	0.14%	0.04%	0.18%	0.07%	0.22%	0.07%	6.9	3
C18:2n6	11.01%	1.44%	22.70%	3.15%	32.19%	4.61%	991.5	259.5
C20:0	0.12%	0.02%	0.17%	0.04%	0.09%	0.03%	2.7	0.8
C18:3n6	0.06%	0.02%	0.31%	0.13%	0.48%	0.18%	14.8	7.1
C20:1n9	0.22%	0.05%	0.26%	0.06%	0.15%	0.05%	4.7	1.6
C18:3n3	0.11%	0.04%	0.39%	0.14%	0.62%	0.22%	19.6	10.9
C20:2n6	0.25%	0.04%	0.23%	0.06%	0.22%	0.04%	6.9	2
C22:0	0.22%	0.06%	0.46%	0.09%	0.16%	0.04%	4.9	1.4
C20:3n6	1.57%	0.37%	1.48%	0.33%	1.40%	0.36%	43.3	15
C20:4n6	15.72%	1.60%	10.97%	1.60%	8.24%	1.93%	252.1	71.6
C24:0	0.42%	0.13%	0.50%	0.12%	0.15%	0.04%	4.5	1.1
C20:5n3	0.96%	0.67%	1.01%	0.72%	1.08%	0.87%	32.3	24.9
C24:1n9	0.35%	0.14%	0.45%	0.14%	0.19%	0.06%	5.7	1.9
C22:4n6	3.36%	0.84%	1.33%	0.34%	0.20%	0.07%	6.2	2.7
C22:5n6	0.57%	0.17%	0.36%	0.10%	0.16%	0.06%	5.1	2.4
C22:5n3	2.97%	0.56%	1.34%	0.30%	0.55%	0.13%	16.9	5.1
C22:6n3	5.19%	1.29%	3.23%	0.92%	2.25%	0.76%	68.4	25
O3I / [FA]	6.15%	1.83%					3094.1	262

Fatty acid nomenclature defined in Table 2.

## Data Availability

The data generated from this study are available upon request to and approval by the authors.

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
