# Peer review of "Derivation of the Omega-3 Index from EPA and DHA Analysis of Dried Blood Spots from Dogs and Cats"

_vetsci, 2022, doi:10.3390/vetsci10010013_

Round 1

Reviewer 1 Report (Previous Reviewer 2)

The content in the manuscript has been improved with the changes made. 

A couple of minor typos:

There is a typo in the title for Table 4: "plasma plasma"

In the supplementary tables, the title for Figure 1 on the first page indicates that these are human samples, but the title (and description) for Figure 1 on page 2 does not mention that these are human samples. 

Author Response

There is a typo in the title for Table 4: "plasma plasma"

FIXED

In the supplementary tables, the title for Figure 1 on the first page indicates that these are human samples, but the title (and description) for Figure 1 on page 2 does not mention that these are human samples.

NOW CORRECTED IN THE TITLE AND IN THE LEGEND

Reviewer 2 Report (New Reviewer)

General: The study aims at validating omega 3 index determination in dogs and cats using dry blood spots samples (DBS). Results were obtained through equations produced from random samples. Omega 3 index were compared with values in humans. DBS sampling works, but the concern with design of the study is is that omega 3 index evaluation is not established for dogs and cats. So, the value of the study is limited to the validation of DBS for fatty acid composition in full blood and the comparison of the fatty acid composition in plasma, red blood cells and full blood (DBS) in dogs, cats and humans.    

The omega 3 index is related to heart disease in humans. Is cardiovascular disease a common problem in cats and dogs? Need to be commented on.

The manuscript has been corrected, which makes it difficult to read in some sections. 

Line 42: Krill oil and fish oil are keywords. Cannot find anything about krill oil and fish oil in the manuscript. Need to be corrected. 

Check the figures, some are duplicates.

The tables should have the sum of plasma fatty acids ug/L. From my calculations fatty acid in dog, cat and human plasma is 5636, 3389 and 3094 ug/mL, respectively.  The plasma fatty acid content is much higher in dogs. This will affect the ratio of fatty acids coming from plasma and RBC membranes. But from the DBS results it appear that they are about equally coming from plasma and RBC. This means that the RBC fatty acid contribution would be much higher in dogs than in humans and cats. But this is given in your report. So, are the RBC smaller and/or at a higher number in dogs that will make fatty acid content higher? 

Author Response

General: The study aims at validating omega 3 index determination in dogs and cats using dry blood spots samples (DBS). Results were obtained through equations produced from random samples. Omega 3 index were compared with values in humans. DBS sampling works, but the concern with design of the study is that omega 3 index evaluation is not established for dogs and cats. So, the value of the study is limited to the validation of DBS for fatty acid composition in full blood and the comparison of the fatty acid composition in plasma, red blood cells and full blood (DBS) in dogs, cats and humans. 

YES IT’S TRUE THAT AT THE MOMENT, THERE ARE NO OPTIMAL OR TARGET LEVELS FOR THE OMEGA-3 INDEX IN DOGS OR CATS. OUR HOPE IS THAT THIS STUDY WILL HELP SET THE STAGE FOR THE DEVELOPMENT OF SUCH TARGET VALUES AS DBS TESTING FOR OMEGA-3 STATUS IS INCLUDED IN FUTURE INTERVENTION STUDIES WHERE ‘HEALTHY’ LEVELS COULD BE DEFINED MORE RIGOROUSLY.    

The omega 3 index is related to heart disease in humans. Is cardiovascular disease a common problem in cats and dogs? Need to be commented on.

THE FOLLOWING HAS BEEN ADDED TO THE CONCLUSIONS SECTION: “IT SEEMS LIKELY THAT OPTIMAL OR TARGET LEVELS WILL BE SPECIES-SPECIFIC (AND POSSIBLY DISEASE-SPECIFIC AS WELL SINCE DOGS AND CATS ARE MORE PRONE TO CHRONIC INFLAMMATORY CONDITIONS THAN TO CARDIOVASCULAR DISEASE WHICH IS MORE COMMON IN HUMANS).

Line 42: Krill oil and fish oil are keywords. Cannot find anything about krill oil and fish oil in the manuscript. Need to be corrected. 

GOOD CATCH. WHEN THE HORSE DATA WERE INCLUDED THEY HAD BEEN FED KRILL OIL. I REOMVED THE HORSES BUT FORGOT ABOUT THE KRILL OIL. IT’S NOW GONE.

Check the figures, some are duplicates.

IN THE VERSION I HAVE I CANNOT FIND ANY. CAN THE REVIEWER BE MORE SPECIFIC?

The tables should have the sum of plasma fatty acids ug/L. From my calculations fatty acid in dog, cat and human plasma is 5636, 3389 and 3094 ug/mL, respectively.  The plasma fatty acid content is much higher in dogs. This will affect the ratio of fatty acids coming from plasma and RBC membranes. But from the DBS results it appear that they are about equally coming from plasma and RBC. This means that the RBC fatty acid contribution would be much higher in dogs than in humans and cats. But this is given in your report. So, are the RBC smaller and/or at a higher number in dogs that will make fatty acid content higher? 

HIGHER PLASMA FATTY ACID CONCENTRATIONS CAN ONLY MEAN HIGHER LEVELS OF CIRCULATING LIPOPROTEINS WHICH APPEARS TO BE THE CASE FROM A QUICK REVIEW OF THE LITERATURE ON DOG AND CAT ‘NORMAL’ LIPID VALUES. THIS COMMENT DOES BRING UP A POINT THAT WE SHOULD HAVE MENTIONED IN OUR LIMITATIONS, THAT IS, THAT WE EXCLUDED FROM THIS STUDY ANY PLASMA SAMPLES THAT WERE VISIBLY LIPEMIC. SUCH SAMPLES CONTAIN A HIGH CONCENTRATION OF CHYLOMICRONS CONSISTENT WITH THE RECENT CONSUMPTION OF A HIGH FAT MEAL. ACCORDINGLY, OUR RESULTS (OUR EQUATIONS) ACTUALLY ONLY APPLY TO FASTING SAMPLES. THIS HAS NOW BEEN ADDED AS ANOTHER LIMITATION AS FOLLOWS:

We also limited this study to plasma samples that were visibly non-lipemic, reflecting the recent consumption of a high fat meal. Since chylomicrons are very rich in triglycerides (i.e., FAs), the DBS omega-3 results would be artificially lower and less translatable to RBC EPA+DHA levels. Hence, our equations should only be used in the fasting state.

This manuscript is a resubmission of an earlier submission. The following is a list of the peer review reports and author responses from that submission.

Round 1

Reviewer 1 Report

The manuscript assigned aimed to develop equations to predict the O3I in dogs, cats and horses from a dried blood spot (DBS) analysis. The method proposed from obtaining blood il less invasive that phlebotomy and this is a good point for the study. On the other hand, the equations derived from the data obtained were valid for dogs, while for cats and horses they were not convincing.

The study, in general, is well exposed and the data well commented in the discussion. However, there are some fundamental issues to highlight:

1)    The number of both horses and cats in the study is low.

The number of horses included in the is really too low and I think that this species should be omitted in the manuscript (if the editors decide to allow it to be considered for publication – see point 2)

The number of cats included in the study is low. Especially compared with the number of dogs which, in fact, gave reliable results.

As stated in the manuscript, both canine and feline samples were derived from a veterinary emergency hospital and the cat is not an unusual patient so, before proposing the study to a journal, a higher quantity of cat samples should have been collected and analyzed. A number of cat comparable with the number of dogs could be the best (around 30).

2)    Conflict of interest. On my opinion the conflicts of interest are relevant and involve too much authors (the first and the last Authors in particular). As the authors state, the first and the second authors hold stock in OmegaQuant Analytics, LLC , the laboratory that performed the sample analyses. In addition. Moreover, the second authors is also an employee of the laboratory. In addition, the last authors,  + two more authors,  are employees of Aker BioMarine Antarctic, AS; the maker of the krill oil product used in the horse study

Considering these issues, I recommend rejection of the manuscript.
